# Efficient Hyperparameter Optimization with Adaptive Fidelity Identification

## Abstract

Hyperparameter optimization is powerful in automatically tuning hyperparameters, with Bayesian Optimization (BO) being a mainstream method for this task. Extending BO into the multi-fidelity setting has been an emerging research topic in this field, but faces the challenge of determining an appropriate fidelity for each hyperparameter configuration to fit the surrogate model. To tackle the challenge, we propose a multi-fidelity BO method named FastBO, which excels in adaptively deciding the fidelity for each configuration and providing strong performance while ensuring efficient resource usage. These advantages are achieved through our proposed techniques based on the concepts of *efficient point* and *saturation point* for each configuration, which can be obtained from the empirical learning curve of the configuration, estimated from early observations. Extensive experiments demonstrate FastBO's superior anytime performance and efficiency in identifying high-quality configurations. We also show that our method provides a way to extend any single-fidelity method to the multi-fidelity setting, highlighting the wide applicability of our approach.

## 1 Introduction

Hyperparameters are crucial in machine learning (ML) pipelines, driving both the efficiency and quality of ML applications. Hyperparameter optimization (HPO) (Feurer & Hutter, 2019) aims to find the hyperparameters for an ML algorithm that can yield good performance without human experts, which is a key topic of automated machine learning (AutoML) (Zöller & Huber, 2021). Among different HPO methods, Bayesian Optimization (BO) (Snoek et al., 2012; Hutter et al., 2011; Bergstra et al., 2011) is an effective model-based method that has shown remarkable success (Dong & Yang, 2020; Siems et al., 2020). BO maintains a *surrogate model* of the target performance metric based on past evaluations of hyperparameter configurations, which guides the choice of more promising configurations to evaluate.

Despite its sample efficiency, standard BO requires a full evaluation of each configuration, involving full-scale training and testing of ML models, which can be highly time-consuming, particularly with the recent trend to larger models. To avoid expensive full evaluations, multi-fidelity methods (Jamieson & Talwalkar, 2016; Li et al., 2017; 2020; Bohdal et al., 2023) have been proposed, where the *fidelities* refer to the levels of performance metrics obtained under different resource levels. These methods follow the principle of successive halving (SHA) (Jamieson & Talwalkar, 2016): initially, they evaluate a set of randomly selected configurations using a small number of resources; then, based on the low-fidelity performances, the poorly-performing ones are successively eliminated, while the well-performing ones continue to be evaluated with progressively increasing resources. Follow-up studies (Falkner et al., 2018; Wang et al., 2018; Klein et al., 2020; Li et al., 2022; Salinas et al., 2023) propose model-based multi-fidelity methods, replacing the random configuration selection with a more informed model to improve sample efficiency.

Nevertheless, the current model-based multi-fidelity methods face a major limitation: they are built upon the SHA framework, which operates under the assumption that learning curves of different configurations rarely intersect. This assumption does not hold in practice (Viering & Loog, 2022), i.e., early performance observations cannot always indicate the final fidelity performance at the full resource level. This leads to a fundamental challenge when extending model-based methods to the multi-fidelity setting: *What is the appropriate fidelity for each configuration to fit the surrogate model?* In other words, which fidelity can provide performance observations that reliably indicate the final fidelity performance? Existing methods struggle to address this fundamental challenge. In particular, BOHB (Falkner et al., 2018) and Hyper-Tune (Li et al., 2022) fit separate surrogate models for different fidelities, failing to capture inter-fidelity correlations. Freeze-Thaw BO (Swersky et al., 2014) and A-BOHB (Klein et al., 2020) fit a joint model but require strong assumptions to remain tractable. Another work by Salinas et al. (2023) suggests using the last

observed fidelity performance to fit the surrogate model. However, it widens the gap between poorly- and well-performing configurations at the early stage, potentially leading to an inaccurate surrogate model.

To this end, we propose a multi-fidelity extension of BO, namely FastBO, which tackles the challenge of determining the appropriate fidelity for each configuration to fit the surrogate model. FastBO identifies a so-called *efficient point* for each configuration to be the fidelity. The point balances computational cost and performance quality while capturing valuable learning curve trends. In essence, FastBO dynamically selects the fidelity for each configuration instead of evaluating all the configurations at the same fidelity. Additionally, FastBO identifies a *saturation point* for each configuration to be an approximation of the final fidelity, leading to high-quality performance while reducing resource wastage. The two crucial points are adaptively derived from the learning curve of each configuration estimated based on early observations. Furthermore, the warm-up and post-processing stages are carefully designed to enable judicious early-termination detection and efficient saturation level evaluation. Our empirical evaluation against the state-of-the-art methods shows that FastBO has strong anytime performance and can considerably save up to 87% of the time required to identify a good configuration. In summary, we make the following major contributions.

1. We propose a multi-fidelity model-based HPO method that can adaptively decide the fidelities for configurations and efficiently offer strong performance, thanks to the introduced concepts of efficient and saturation points.

2. We develop the learning curve modeling technique to enable adaptive derivation of the two key points, a warm-up stage to allow early-termination detection, and a post-processing stage to ensure efficient saturation level evaluation.

3. We show that our strategy can be used to extend existing single-fidelity methods to the multi-fidelity setting, demonstrating the effectiveness and generality of our method and highlighting promising future opportunities.

## 2 RELATED WORK

The rising costs of evaluating machine learning models have made it intractable to use simple methods like random search (Bergstra & Bengio, 2012) to find suitable hyperparameter configurations within a reasonable amount of time. Two crucial directions to efficiently solve the HPO problem are model-based methods and multi-fidelity methods, both of which are highly relevant to our work. Ideas from these two directions can also be combined.

**Model-based methods.** Bayesian Optimization (BO) stands as the representative of model-based methods. Based on the *surrogate model* constructed by historical evaluation results, BO selects the next configuration to evaluate via an *acquisition function* that balances exploration and exploitation. Commonly used surrogate models are Gaussian processes (Snoek et al., 2012), random forests (Hutter et al., 2011), the tree-structured Parzen estimator (TPE) (Bergstra et al., 2011), and deep neural networks (Snoek et al., 2015; Springenberg et al., 2016). Popular choices of the acquisition function include Expected Improvement (Mockus, 1998), Upper Confidence Bound (Srinivas et al., 2009), Entropy Search (Hennig & Schuler, 2012), and Predictive Entropy Search (Hernández-Lobato et al., 2014). Recent studies on BO have explored the utilization of expert priors (Shahriari et al., 2016; Oh et al., 2018; Li et al., 2018; Hvarfner et al., 2022) and derivative information (Wu et al., 2017; Padidar et al., 2021; Ament & Gomes, 2022).

**Multi-fidelity methods.** Multi-fidelity methods exploit low and high fidelities for configuration evaluations in order to save the evaluation time. Different fidelities correspond to different resource levels, typically training epochs or training subset ratios. Successive halving algorithm (SHA) (Jamieson & Talwalkar, 2016) runs a set of hyperparameter configurations using a small number of resources and then successively promotes only the best-performing half of configurations to continue for twice as many resources. Hyperband (Li et al., 2017) calls SHA as a sub-routine with varying maximum resources for a single configuration and introduces a reduction factor to control the fraction of configuration promotion. ASHA (Li et al., 2020) extends SHA to the asynchronous setting by aggressive early-stopping. Later, PAHSA (Bohdal et al., 2023) further extends ASHA through more aggressive early-stopping according to the ranking of configurations during the tuning procedure.

**Combination of model-based and multi-fidelity methods.** Several studies propose to combine the model-based and multi-fidelity methods to leverage the advantages of both. BOHB (Falkner et al., 2018) and a parallel work (Wang et al., 2018) first combine BO and Hyperband by replacing the random sampling in Hyperband with TPE-based BO. A-BOHB (Klein et al., 2020) employs a joint GP surrogate model over fidelities and supports asynchronous scheduling. Hyper-Tune (Li et al., 2022) improves its Hyperband component by D-ASHA, which is a delayed strategy to decrease inaccurate promotions. Salinas et al. (2023) proposed to extend single-fidelity methods to multi-fidelity settings by using the performance of the last fidelity in a standard ASHA running.

## 3 Problem Formulation

Given a machine learning algorithm having hyperparameters $\lambda_1, ..., \lambda_m$ with respective domains $\Lambda_1, ..., \Lambda_m$, we define its hyperparameter space as $\Lambda = \Lambda_1 \times ... \times \Lambda_m$. Here, we formally define the problem and outline the key challenge related to hyperparameter optimization (HPO). All the notations used are summarized in Appendix A.1 for reference.

**Single-fidelity setting.** For each hyperparameter configuration $\lambda$, we denote $f(\lambda)$ as the performance metric achieved using $\lambda$. For consistency, the metric in this paper refers to descending metrics like validation loss, with ascending metrics being treated similarly. In the single-fidelity HPO setting, we aim to find $\lambda^*$ minimizing the function $f(\lambda)$:

$$\lambda^* = \underset{\lambda \in \Lambda}{\arg\min}\, f(\lambda). \tag{1}$$

Bayesian Optimization (BO) is arguably one of the most popular approaches to solve the HPO problem. The vanilla BO has two key components: a probabilistic *surrogate model* to approximate the objective function $f(\lambda)$, and an *acquisition function* to identify a promising configuration from search space that can trade-off exploration and exploitation. With these ingredients, BO iterates the following three steps: (i) select the most promising configuration $\lambda_i$ by maximizing the acquisition function; (ii) evaluate the configuration $\lambda_i$ to get its performance $y_i$ and add the resulting data $(\lambda_i, y_i)$ into the current observation set $\mathcal{D}_{i-1} = \{(\lambda_1, y_1), ..., (\lambda_{i-1}, y_{i-1})\}$; (iii) update the surrogate model and the acquisition function based on the augmented $\mathcal{D}_i$.

**Multi-fidelity setting.** Multi-fidelity HPO methods consider additional resource information, such as training epochs or training subset ratios. Evaluating configurations at various resource levels results in different performance levels, known as the *fidelities*. Different fidelities provide a way to balance computational cost and performance quality. In the multi-fidelity setting, the target optimization problem in Equation 1 is extended to $\lambda^* = \underset{\lambda \in \Lambda}{\arg\min}\, f(\lambda, r_{max})$, where $f(\lambda, r_{max})$ is the objective function obtained for configuration $\lambda$ at the maximum resource level $r_{max}$. We use $r$ to denote the resource level, which can also be interpreted as the fidelity, and $r \in \{r_{min}, ..., r_{max}\}$.

**Extending single-fidelity methods to the multi-fidelity setting.** The inefficiency of single-fidelity methods stems from their reliance on the final fidelity evaluation of $f(\lambda, r_{max})$ to fit a surrogate model, which incurs high cost due to the full evaluation of the configurations. Notably, low-fidelity evaluations of $f(\lambda, r)$ for $r < r_{max}$ provide informative insights into $f(\lambda, r_{max})$ but are computationally cheaper, which are valuable to the optimization process. Therefore, we seek an effective way to extend single-fidelity methods like BO to the multi-fidelity setting. More specifically, recalling the earlier steps of BO, when evaluating the configuration $\lambda_i$ in the second step, we instead acquire its low-fidelity performance $y_i^{r_i}$ at $r_i$, where $r_i$ denotes the fidelity used for $\lambda_i$ to fit the surrogate model. The observations $\mathcal{D}_i$ then becomes $\{(\lambda_1, y_1^{r_1}), ..., (\lambda_i, y_i^{r_i})\}$. To conclude, in order to extend single-fidelity methods to the multi-fidelity setting, the key challenge to be addressed is to determine $r_i$ for each $\lambda_i$.

## 4 Methodology

In this section, we propose a novel multi-fidelity model-based algorithm FastBO. We first propose the key concepts of efficient point and saturation point, which are crucial in deciding the fidelity level to fit the surrogate model and to approximate the final fidelity respectively. Secondly, we elaborate on the details of learning curve modeling, where the two crucial points can be extracted. Then, we present the techniques associated with the auxiliary warm-up and post-processing stages. Finally, we summarize FastBO and discuss its wide applicability to any single-fidelity methods.

### 4.1 Adaptive Estimation of Efficient and Saturation Points

In our method, we adaptively identify two pivotal points for each configuration $\lambda_i$: the *efficient point* and the *saturation point*, which are crucial in the optimization process. We first formally define the efficient point as follows.

**Definition 1** (Efficient point). *For a given learning curve $\mathcal{C}_i(r)$ of hyperparameter configuration $\lambda_i$, where $r$ represents the resource level (also referred to as fidelity), the efficient point $e_i$ of $\lambda_i$ is defined as: $e_i = \min\{r \mid \mathcal{C}_i(r) - \mathcal{C}_i(2r) < \delta_1\}$, where $\delta_1$ is a predefined small threshold.*

The semantic of Definition 1 is that starting from the efficient point onwards, when the resources are doubled (i.e., from $r$ to $2r$), the performance improvement falls below a small threshold $\delta_1$. Consequently, this point characterizes the fidelity at which a configuration demonstrates strong performance while still efficiently utilizing resources. In simpler terms, it signifies an appropriate fidelity of performance that can be achieved with comparably efficient resource usage. Building upon the above definition, we make the following remark.

**Remark 1.** *The efficient points of the hyperparameter configurations can serve as their appropriate fidelities used for fitting the surrogate model. This is due to their (i) optimal resource-to-performance balance, (ii) ability to capture valuable learning curve trends, and (iii) customization for different hyperparameter configurations.*

We elaborate on the reasons in Remark 1 as follows. Firstly, efficient points balance the trade-off between computational cost and result quality. Beyond the efficient point of a given configuration, allocating additional resources to that configuration becomes less efficient. Secondly, efficient points capture valuable trends within the learning curves. For example, the learning rate influences the shape of learning curves; the identification of efficient points for configurations with smaller learning rates often occurs at later stages. The insights into learning curve behaviors enable more informed decision-making. Thirdly, the ability to customize the fidelity for each specific configuration is a significant advantage. This adaptive approach is more reasonable than previous studies that use a fixed fidelity for all configurations, as it better accounts for the unique characteristics of each learning curve.

This insight leads us to leverage the efficient point $e_i$ identified for each configuration $\boldsymbol{\lambda}_i$ as its fidelity used to fit the surrogate model. More specifically, we evaluate $\boldsymbol{\lambda}_i$ until reaching $e_i$ and obtain the observed performance $y_i^{e_i}$. The resulting data point $(\boldsymbol{\lambda}_i, y_i^{e_i})$ is then added into the current observation set $\mathcal{D}_{i-1}$ to refit the surrogate model.

In addition to efficient points, we identify saturation points for all configurations from their learning curves as well. We provide the formal definition of the saturation point as follows.

**Definition 2** (Saturation point). *For a given learning curve $\mathcal{C}_i(r)$ of hyperparameter configuration $\boldsymbol{\lambda}_i$, where $r$ represents the resource level (also referred to as fidelity), the saturation point $s_i$ of $\boldsymbol{\lambda}_i$ is defined as: $s_i = \min\{r \mid \forall r' > r, |\mathcal{C}_i(r') - \mathcal{C}_i(r)| < \delta_2\}$, where $\delta_2$ is a predefined small threshold.*

The semantic of Definition 2 is that beyond the saturation point, the observed performance no longer exhibits notable variations with more resources. Consequently, this point characterizes the fidelity at which the performance of a configuration stabilizes. The concept of saturation point is well-recognized within the machine learning community. Building upon the above definition, we make the following remark.

**Remark 2.** *The saturation points of the hyperparameter configurations can serve as their approximate final fidelities, as they provide performance results that meet predefined quality thresholds while reducing resource wastage.*

This insight leads us to use the saturation point $s_i$ identified for each configuration $\boldsymbol{\lambda}_i$ as its final fidelity approximation. The point is used in the post-processing stage for promoting some well-performing configurations to get higher-fidelity performances. In essence, when aiming for a full evaluation of the configurations, we suggest that terminating the evaluation at the saturation point is sufficient. A more intuitive illustration of the concepts is provided in Appendix A.2.

## 4.2 LEARNING CURVE MODELING

From Definitions 1 and 2, we can extract the efficient points and saturation points of the configurations from their respective learning curves. The learning curve $\mathcal{C}_i(r)$ corresponds to hyperparameter configuration $\boldsymbol{\lambda}_i$ and describes the predictive performance with $\boldsymbol{\lambda}_i$ as a function of the fidelity $r$. Here, $r$ can be either the number of training instances or the number of training epochs or iterations. In the context of learning curves, the former is referred to as observation learning curves, while the latter is iteration learning curves (Mohr & van Rijn, 2022). Both types are applicable to FastBO, so we use the term learning curve to encompass both. Given the observation set $\mathcal{O}_i^w = \{(r, y_i^r)\}_{r=r_{min},...,w}$ for configuration $\boldsymbol{\lambda}_i$, which comprises pairs of data points representing fidelities $r \in \{r_{min}, ..., w\}$ and the corresponding evaluations $y_i^r$, FastBO can estimate a learning curve for $\boldsymbol{\lambda}_i$ based on $\mathcal{O}_i^w$ by first constructing a parametric learning curve model, then estimating parameters in the model.

**Constructing a parametric learning curve model.** Empirical learning curves can be modeled with function classes relying on some parameters. Viering & Loog (2022) comprehensively summarized the parametric models studied in machine learning. In practice, different problems have different learning curves; even under the same problem, different hyperparameter configurations (e.g., learning rate, regularization, etc.) may lead to significantly different learning curves. Since one single parametric model is not enough to characterize all the learning curves by itself, we consider combining different parametric models into a single model. Specifically, we consider three parametric models POW3, EXP3 and LOG2, as listed in Table 1, which have shown good fitting and predicting performance in

Table 1: Parametric learning curve models used.

| Model | Formula | Family |
|---|---|---|
| POW3 | $y = d + ax^{-\alpha}$ | Power law |
| EXP3 | $y = d + e^{-ax+b}$ | Exponential |
| LOG2 | $y = d + a\log(x)$ | Logarithmic |

previous empirical studies (Viering & Loog, 2022; Mohr & van Rijn, 2022). Sigmoidal models like MMF and Weibull are not being considered, since they tend to fit well if enough observations are used for fitting; but in situations like ours where observations are limited, their performance is suboptimal (Mohr et al., 2022).

Here, we denote each of the three parametric models as $c_j(r|\boldsymbol{\theta}_j)$ with parameters $\boldsymbol{\theta}_j$, where the independent variable $r$ represents the fidelity. We combine three models into one model through a weighted linear combination:

$$\mathcal{C}(r|\boldsymbol{\phi}) = \sum_{j \in \{1,2,3\}} \omega_j c_j(r|\boldsymbol{\theta}_j), \quad \boldsymbol{\phi} = \{\omega_1, \omega_2, \omega_3, \boldsymbol{\theta}_1, \boldsymbol{\theta}_2, \boldsymbol{\theta}_3\}, \tag{2}$$

where $\boldsymbol{\phi}$ is the parameter of the combined model, which consists of parameters $\{\boldsymbol{\theta}_1, \boldsymbol{\theta}_2, \boldsymbol{\theta}_3\}$ and weight $\{\omega_1, \omega_2, \omega_3\}$ of every single model. Therefore, each pair of observations $(r, y_i^r)$ in $O_i^w$ can be modeled by the combined parametric model as $y_i^r = \mathcal{C}(r|\boldsymbol{\phi}) + \epsilon$, where $y_i^r$ is the observed dependent variable and $\epsilon$ represents the error term.

**Estimating parameters in the parametric learning curve model.** We employ maximum likelihood estimation to estimate the parameters $\boldsymbol{\phi}$ in the parametric model $\mathcal{C}(r|\boldsymbol{\phi})$. Assuming that $\epsilon \sim \mathcal{N}(0, \sigma^2)$, the probability of an observed performance $y_i^r$ under parameters is given by $p(y_i^r|\boldsymbol{\phi}, \sigma^2) = \mathcal{N}(y_i^r; \mathcal{C}(r|\boldsymbol{\phi}), \sigma^2)$. Given the observations $\mathcal{O}_i^w$ of $\boldsymbol{\lambda}_i$ that contains a set of observed data points $(r, y_i^r)$, the likelihood function can be expressed as:

$$\mathcal{L}(\boldsymbol{\phi}, \sigma^2; \mathbf{r}, \mathbf{y}_i^r) = \prod p(y_i^k|\boldsymbol{\phi}, \sigma^2) = \prod_{k=r_{min}}^{w} \frac{1}{\sigma\sqrt{2\pi}} \exp\left(-\frac{(y_i^r - \mathcal{C}(r=k|\boldsymbol{\phi}))^2}{2\sigma^2}\right). \tag{3}$$

We find the parameters $\boldsymbol{\phi}$ by maximizing the log-likelihood function, which can be easily calculated given Equation 3.

An existing model-free method (Domhan et al., 2015) also considers using learning curves for the HPO problem. However, it targets predicting the high-fidelity performance from the low-fidelity observations and thus stopping configurations that are unlikely to beat the current best values, which is different from our main target of identifying appropriate fidelity levels for the configurations to fit the surrogate model from their estimated learning curves.

### 4.3 AUXILIARY WARM-UP AND POST-PROCESSING STAGES

In addition to its core components, FastBO incorporates two auxiliary stages: the warm-up and post-processing stages. For the completeness of our method, we provide an overview of these stages, outlining their targets and presenting the key techniques of early-termination detection and saturation-level evaluation that are applied within.

**Warm-up stage.** The warm-up stage prepares the early observation set $\mathcal{O}_i^w$ for each configuration $\boldsymbol{\lambda}_i$ that is used to estimate its learning curve, as discussed in § 4.2. Here $w \in (r_{min}, r_{max})$ a pre-determined fidelity, denoted as warm-up point. Specifically, we initiate the evaluation of each newly selected $\boldsymbol{\lambda}_i$, proceeding until reaching $w$. During this process, we record each fidelity $r$ and its corresponding evaluation result $y_i^r$, forming pairs $(r, y_i^r)$. Upon reaching $w$, we pause the evaluation for $\boldsymbol{\lambda}_i$ and obtain its early observation set $\mathcal{O}_i^w = \{(r, y_i^r)\}_{r=r_{min},...,w}$, and thus start modeling the learning curve. During the warm-up stage, we monitor the performance changes across every two continuous fidelities. If we detect that the performance of a configuration $\boldsymbol{\lambda}_i$ has consecutively dropped twice by more than a ratio $\alpha$, i.e., $(y_i^{r-1} - y_i^{r-2}) > \alpha y_i^{r-2}$ and $(y_i^r - y_i^{r-1}) > \alpha y_i^{r-1}$, we promptly terminate the evaluation for $\boldsymbol{\lambda}_i$ at its current fidelity $r$, because such consecutive performance deterioration indicates that $\boldsymbol{\lambda}_i$ is unlikely to achieve satisfactory performance. Once terminated, we directly incorporate the current performance $y_i^r$ of $\boldsymbol{\lambda}_i$ into the observation set $\mathcal{D}_{i-1}$ that is used for updating the surrogate model. Thus, further operations like learning curve modeling are discontinued for $\boldsymbol{\lambda}_i$. Moreover, if we observe a single case of performance drop without subsequent occurrences, i.e, $y_i^{r-1} - y_i^{r-2} > \alpha y_i^{r-2}$ and $y_i^r - y_i^{r-1} \leq \alpha y_i^{r-1}$, we opt not to include data from fidelity $r-1$ in $\mathcal{O}_i^w$. This is to manually filter out potential noise in the data that may adversely affect the fitting of the learning curve.

**Post-processing stage.** The post-processing stage aims at two tasks: promoting the well-performing configurations for saturation-level evaluations and identifying the best configuration and its performance. Firstly, FastBO promotes the top-$k$ well-performing configurations and evaluates them to their saturation points to ensure high-quality performance while maintaining efficient resource utilization. We set $k$ to be always less than or equal to the number of parallel workers available, ensuring a manageable overhead of saturation-level evaluations. It is worth noting that the additional time required is factored into the overall time. Secondly, FastBO finds the best configuration along with its performance achieved so far, which is a standard final step in most HPO methods. However, we observe that an increase in fidelities does not always result in performance improvement. Factors such as overfitting, resource saturation, or problem complexity can contribute to this phenomenon. Therefore, we treat the evaluation at each fidelity as an individual task, recording all these intermediate evaluation results. In this way, FastBO finds the best performance

by considering all the results, rather than relying solely on the highest-fidelity performances of the configurations. In the parallel setting, treating each fidelity evaluation as an individual task offers an added benefit due to its finer granularity. More specifically, when a worker is idle, it takes on a new task of evaluating a configuration at a specific fidelity, rather than evaluating an entire configuration.

## 4.4 FASTBO AND GENERALIZATION

Algorithm 1 summarizes our proposed FastBO. It takes surrogate model $\mathcal{M}$, acquisition function $a$, warm-up point $w$, performance decrease ratio $\alpha$, promotion number $k$, and thresholds $\delta_1$, $\delta_2$ as inputs, and output the best-founded configuration $\boldsymbol{\lambda}^*$ and its performance $y^*$. FastBO follows a similar iterative process of model-based methods but replaces the expensive full evaluations with a more intelligent and efficient alternative (cf. Lines 4-10). Specifically, each new configuration $\boldsymbol{\lambda}_i$ first enters a warm-up stage to collect its early observation set $\mathcal{O}_i^w$ and to be detected and terminated if it exhibits consecutive performance deterioration (cf. Line 4). If $\boldsymbol{\lambda}_i$ is not terminated, FastBO then estimates a learning curve $\mathcal{C}_i(r)$ for $\boldsymbol{\lambda}_i$ based on $\mathcal{O}_i^w$ (cf. Line 6), and thus the efficient point and saturation point of $\boldsymbol{\lambda}_i$ can be adaptively obtained (cf. Line 7). After that, $\boldsymbol{\lambda}_i$ continues to be evaluated until reaching $e_i$ (cf. Line 8); the result is added to the observation set $\mathcal{D}$ (cf. Line 11) that is used for updating $\mathcal{M}$ (cf. Line 12). On the other hand, the

---

**Algorithm 1:** FastBO algorithm

**input** : $\mathcal{M}, a, w, \alpha, k, \delta_1, \delta_2$.
**output:** $\boldsymbol{\lambda}^*, y^*$
1   $i \leftarrow 0, \mathcal{D} \leftarrow \emptyset$
2   **while** *not meet the stop criterion* **do**
3      find $\boldsymbol{\lambda}_i \leftarrow \arg\max_{\boldsymbol{\lambda} \in \boldsymbol{\Lambda}} a(\boldsymbol{\lambda}, \mathcal{M}_{i-1})$
4      $\mathcal{O}_i^w, t \leftarrow$ warm-up given $w, \alpha$ // cf. §4.3
5      **if** $\mathcal{O}_i^w$ *is not empty* **then**
6         fit $\mathcal{C}_i(r)$ to $\mathcal{O}_i^w$ for $\boldsymbol{\lambda}_i$ // cf. §4.2
7         find $e_i, s_i$ of $\boldsymbol{\lambda}_i$ given $\mathcal{C}_i(r), \delta_1, \delta_2$ // cf. §4.1
8         $y_i^{e_i} \leftarrow$ continue evaluating $\boldsymbol{\lambda}_i$ to $e_i$
9      **else**
10         $e_i \leftarrow t, s_i \leftarrow r_{max}$
11      $\mathcal{D}_i \leftarrow \mathcal{D}_{i-1} \cap (\boldsymbol{\lambda}_i, y_i^{e_i})$
12      refit $\mathcal{M}_i$ to $\mathcal{D}_i$
13      $i \leftarrow i + 1$
14   $\boldsymbol{\lambda}^*, y^* \leftarrow$ post-process given $\mathbf{s} = \{s_i\}, k$ // cf. §4.3

---

poorly-performing configuration will be terminated early at fidelity $t$ with its result being added directly to $\mathcal{D}$ (cf. Lines 10, 11). Finally, the post-processing stage promotes the most promising configurations to their saturation points and finds the best-founded configuration $\boldsymbol{\lambda}^*$ and its performance $y^*$ (Line 14).

**Generalizing FastBO to single-fidelity methods.** The core of FastBO is to tackle the key challenge of determining an appropriate fidelity for each configuration to fit the surrogate model by adaptively identifying its efficient point. This adaptive strategy of using the efficient point performances of configurations for surrogate model fitting also provides a simple but effective way to bridge the gap between single- and multi-fidelity methods. While it is primarily described in the context of model-based methods, the strategy can be generalized to various single-fidelity methods. For example, when evaluating the configurations within the population for an evolutionary algorithm-based HPO method, we can similarly evaluate the efficient point performances instead of the final performances of these configurations and integrate the performances in the subsequent processes, such as selection and variation. Relying on the efficient point of each configuration rather than the final fidelity or all fidelities available simplifies the extension of the single-fidelity methods to the multi-fidelity setting. The rationale behind this adaptive fidelity identification strategy is discussed in Remark 1. We also demonstrate in our experiments the efficacy of this strategy in extending a range of single-fidelity methods to the multi-fidelity setting.

## 5 EXPERIMENTS

We empirically evaluate the performance of FastBO and compare it with the random search baseline (RS) and 8 competitive baselines from 3 most related categories, including (i) model-based methods: standard Gaussian Process-based Bayesian Optimization (BO) (Snoek et al., 2012); (ii) multi-fidelity methods: ASHA (Li et al., 2020), Hyperband (Li et al., 2017), PASHA (Bohdal et al., 2023); and (iii) model-based multi-fidelity methods: A-BOHB (Klein et al., 2020), A-CQR (Salinas et al., 2023), BOHB (Falkner et al., 2018), Hyper-Tune (Li et al., 2022). Among these baselines, RS and BO are single-fidelity baselines, while the others are multi-fidelity baselines. Our experiments are conducted on 10 datasets, coming from 3 commonly-used benchmarks including LCBench (Zimmer et al., 2021), NAS-Bench-201 (Dong & Yang, 2020) and FCNet (Klein & Hutter, 2019) that have 7, 6, and 9 hyperparameters, respectively. Detailed information on the benchmarks is provided in Appendix A.5.1.

All the experiments are evaluated with four workers and are repeated using 10 random seeds. FastBO uses a Matérn $\frac{5}{2}$ kernel with automatic relevance determination parameters and the expected improvement acquisition function. We

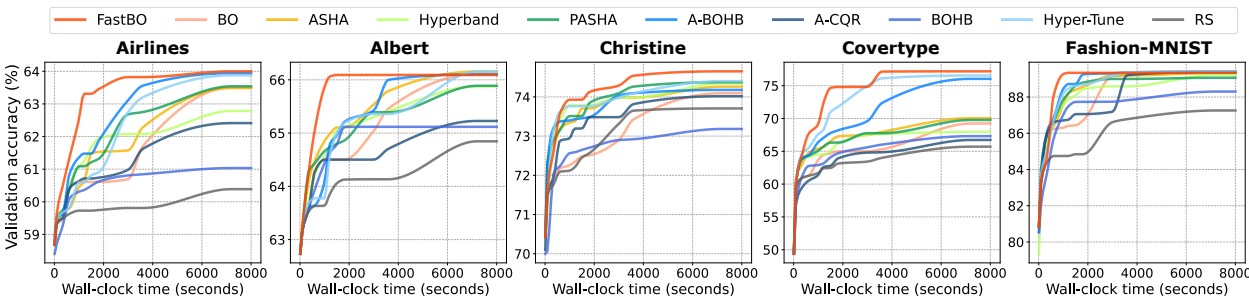

Figure 1: Performance of average validation accuracy on the LCBench benchmark.

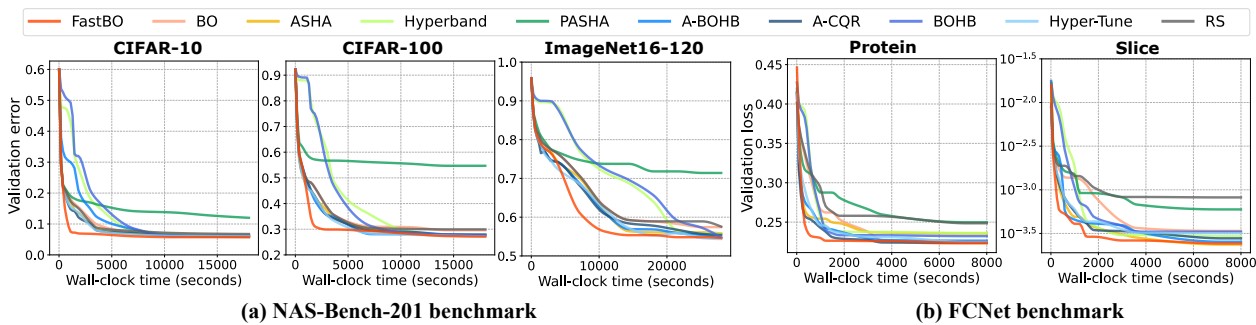

**(a) NAS-Bench-201 benchmark**            **(b) FCNet benchmark**

Figure 2: Performance of **(a)** average validation error on NAS-Bench-201 and **(b)** average validation loss on FCNet.

allocate 20% total resource budget to the warm-up stage, i.e., $w = r_{min} + 0.2 \cdot (r_{max} - r_{min})$. Ratio $\alpha$ is set to 0.1; thresholds $\delta_1$ and $\delta_2$ are set to 0.001 and 0.0005 [1]. We set $k$ according to the number of workers $\#workers$ and the number of started configurations $\#configs$: $k = \max\{\lceil\#configs/10\rceil, \#workers\}$. We use implementations of the baselines provided in Syne Tune (Salinas et al., 2022). Further details of the baseline settings are in Appendix A.5.2.

## 5.1 ANYTIME PERFORMANCE

To evaluate the anytime performance, we compare FastBO against the baselines on wall-clock time. For fair comparisons, all the baselines, even single-fidelity methods BO and RS, are extended to consider intermediate results at all the fidelities, akin to FastBO as discussed in § 4.3. Consequently, all the baselines identify the best configuration from all the intermediate results and thus are able to achieve their best possible anytime performance.

The results on the LCBench, NAS-Bench-201, and FCNet benchmarks are shown in Figures 1 and 2. We report the validation accuracy, validation error, and validation loss over wall-clock time for the three benchmarks, as provided by the benchmarks. Overall, FastBO can handle various performance metrics and shows strong anytime performance for all the datasets. We can observe that FastBO gains an advantage earlier than other methods, rapidly converging to the global optimum after the initial phase. The superiority can be attributed to two main factors. Firstly, FastBO maintains, and in some cases even surpasses, the sample efficiency of vanilla BO, thanks to our techniques that enable quick and precise identification of the fidelities for configurations to update the surrogate model. We provide more explanations and conduct more experiments on sample efficiency in Appendix A.3. Secondly, the multi-fidelity extension speeds up the evaluation for each configuration, contributing to its overall efficiency. In contrast, the single-fidelity baselines tend to waste more time on the full evaluation of the configurations. While the multi-fidelity baselines efficiently explore numerous configurations, they limit their evaluations to only constrained resource levels for some time, thus struggling to provide relatively high performance in a short time. This issue in multi-fidelity methods is particularly pronounced in the PASHA algorithm when applied to NAS-Bench-201 and FCNet, as shown in Figure 2. It is worth noting that the additional computational overhead introduced by FastBO is taken into account in the wall-clock time.

Regarding the final performance, most methods are able to converge to satisfactory solutions, with negligible differences among them in most cases. Although our goal is not to offer the best final performance as we limit the

---

[1]Parameters $\delta_1$ and $\delta_2$ given here are derived after standardizing the various performance metrics to a uniform scale from 0 to 1.

Table 2: Comparison of relative efficiency on configuration identification. Wall-clock time (abbr. WC time) reports the elapsed time spent for each method on finding configurations with similar performance metrics, i.e., validation error for Covertype and ImageNet16-120 ($\times 10^{-2}$) and validation loss for Slice ($\times 10^{-5}$). Regarding relative efficiency, FastBO is set as the baseline with a relative efficiency of 1.00.

| Metric \ Method Dataset | | **FastBO** | BO | PASHA | A-BOHB | A-CQR | BOHB | Hyper-Tune |
|---|---|---|---|---|---|---|---|---|
| Covertype | Val. error | $\mathbf{22.9_{\pm 0.2}}$ | $23.0_{\pm 0.3}$ | $25.1_{\pm 2.5}$ | $23.5_{\pm 1.1}$ | $31.6_{\pm 1.9}$ | $32.5_{\pm 0.8}$ | $23.0_{\pm 0.2}$ |
| | WC time (h) | $\mathbf{0.7_{\pm 0.3}}$ | $2.9_{\pm 0.7}$ | $3.9_{\pm 1.0}$ | $2.0_{\pm 1.0}$ | $3.9_{\pm 0.2}$ | $2.5_{\pm 1.0}$ | $1.8_{\pm 0.7}$ |
| | Rel. efficiency | **1.00** | 0.25 | 0.18 | 0.37 | 0.19 | 0.29 | 0.40 |
| ImageNet 16-120 | Val. error | $\mathbf{55.3_{\pm 0.2}}$ | $57.4_{\pm 1.2}$ | $55.7_{\pm 0.3}$ | $55.8_{\pm 1.6}$ | $55.5_{\pm 0.9}$ | $55.5_{\pm 1.1}$ | $55.3_{\pm 2.0}$ |
| | WC time (h) | $\mathbf{2.2_{\pm 0.7}}$ | $6.6_{\pm 0.9}$ | $2.5_{\pm 1.2}$ | $5.9_{\pm 1.1}$ | $6.0_{\pm 1.3}$ | $3.2_{\pm 0.7}$ | $3.4_{\pm 1.1}$ |
| | Rel. efficiency | **1.00** | 0.34 | 0.90 | 0.38 | 0.37 | 0.68 | 0.67 |
| Slice | Val. loss | $\mathbf{26.3_{\pm 2.6}}$ | $26.4_{\pm 4.4}$ | $26.8_{\pm 9.5}$ | $26.3_{\pm 6.3}$ | $27.1_{\pm 4.2}$ | $26.8_{\pm 5.6}$ | $28.7_{\pm 1.3}$ |
| | WC time (h) | $\mathbf{0.4_{\pm 0.1}}$ | $3.1_{\pm 0.7}$ | $1.2_{\pm 0.9}$ | $2.1_{\pm 0.7}$ | $2.5_{\pm 0.7}$ | $2.2_{\pm 0.9}$ | $1.8_{\pm 0.6}$ |
| | Rel. efficiency | **1.00** | 0.13 | 0.35 | 0.20 | 0.17 | 0.19 | 0.24 |

evaluations to at most the saturation point even for those we consider most promising, FastBO still achieves top-2 final performance on 8 out of 10 datasets. In contrast, model-free methods sometimes cannot obtain a satisfactory final performance because they randomly select the configurations. For example, on the "Covertype" dataset, only 3 out of 2000 configurations yield a validation accuracy exceeding 75%. As a result, all the model-free methods face challenges in converging to a satisfactory final performance.

## 5.2 EFFICIENCY ON CONFIGURATION IDENTIFICATION

One explanation for PASHA's suboptimal anytime performance, as shown in Figure 2, lies in its primary goal (Bohdal et al., 2023): the goal of PASHA is not high accuracy but to identify the best configuration more quickly. To ensure equitable comparisons, we report the time spent for each HPO method on identifying a satisfactory configuration, consistent with the experiments described in Bohdal et al. (2023). Results on three expensive datasets "Covertype"[2], "ImageNet16-120", and "Slice" of the three benchmarks are shown in Table 2. Similar results on additional datasets can be found in Appendix A.4.1. Besides PASHA, results of other model-free multi-fidelity methods are not included, as PASHA demonstrates its superiority over them.

Table 2 shows that FastBO saves 10% to 87% wall-clock time over other methods when achieving up to 9.6% better performance values. It can be observed from the "rel. efficiency" rows, where we set FastBO as the baseline with a relative efficiency of 1.00 and report the efficiency of other methods relative to ours. When compared with vanilla BO, FastBO significantly shortens the time in identifying a good configuration by a factor of 3 to 8, because FastBO can pause a configuration earlier at an appropriate fidelity and fit the surrogate model to guide the next configuration search. This advantage creates opportunities to efficiently explore more configurations, leading to high efficiency in identifying good configurations. Another observation is that PASHA always gets a relatively high variance in wall-clock time. This is due to the fact that different random seeds can have a larger impact on such model-free methods.

## 5.3 EFFECTIVENESS OF ADAPTIVE FIDELITY IDENTIFICATION

As discussed in § 4.1, FastBO is able to adaptively identify the efficient point $e_i$ for each configuration $\boldsymbol{\lambda}_i$ and serves $e_i$ as its fidelity $r_i$ for surrogate model fitting. To investigate how the adaptive fidelity identification strategy impacts the optimization process, we conduct an ablation study to evaluate the performance achieved with and without applying this strategy. Specifically, we compare FastBO, where $r_i$ is adaptively set to $e_i$, with the partial evaluation schemes that employ fixed predefined values as the fidelity for all the configurations to fit the surrogate model. We consider three representative fixed fidelities, including 25%, 50%, and 75% of the total resource budget. In addition, we include a comparison with vanilla BO that can be viewed as using 100% resource budget as the fixed fidelity for all configurations for surrogate model fitting. We provide the results on three representative datasets of the benchmarks in

---

[2]We convert the validation accuracy of "Covertype" into validation error for better readability.

Figure 3, with more results available in Appendix A.4.2. We have three main observations. Firstly, FastBO always outperforms the partial evaluation baselines that use a fixed fidelity for all the configurations, indicating the effectiveness of the adaptive fidelity identification strategy. Secondly, compared to the vanilla BO, partial evaluation schemes with fixed $r_i$ converge faster in the initial stage due to their ability to evaluate more configurations promptly. However, this advantage is gradually

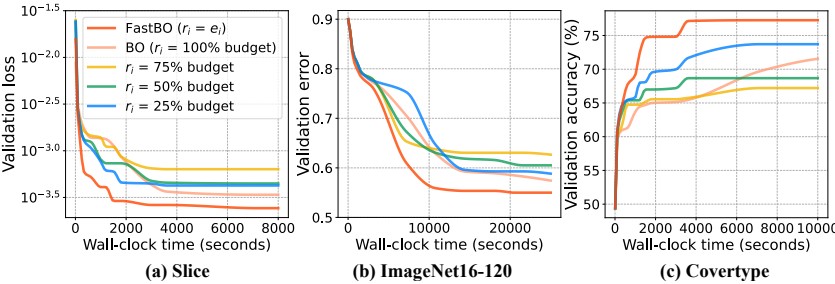

**(a) Slice**   **(b) ImageNet16-120**   **(c) Covertype**

Figure 3: Performance of (i) FastBO that adaptively sets $r_i = e_i$, (ii) the schemes that use fixed 25%, 50%, 75% of the total resource budget as $r_i$ for all configurations, and (ii) vanilla BO that uses 100% total resource budget as $r_i$.

offset over time because they fail to find appropriate fidelities to create an accurate surrogate model. This results in a suboptimal final performance compared to vanilla BO, as shown in Figures 3(a) and 3(b). In the case of Figure 3(c), we can observe a noticeable upward trend exhibited by the vanilla BO towards the end of the evaluation, indicating its potential to improve the final performance given abundant time. The comparison between the partial evaluation baselines and vanilla BO also demonstrates the importance of our adaptive strategy, which ensures that the fidelities align optimally with each configuration. Thirdly, FastBO shows stronger performance than vanilla BO. The superiority of FastBO is due to its good sample efficiency and its fast evaluation of each configuration, as discussed in § 5.1. The limitation of vanilla BO lies in the additional time required for full evaluations.

## 5.4 GENERALITY OF THE PROPOSED EXTENSION METHOD

The adaptive fidelity identification strategy provides a simple way to extend single-fidelity methods to the multi-fidelity setting, as discussed in § 4.4. To examine the ability of our adaptive strategy as an extension method, we conduct experiments using three popular single-fidelity methods CQR (Salinas et al., 2023), BORE (Tiao et al., 2021) and REA (Real et al., 2019), extending them to the multi-fidelity variants with our extension method, referred

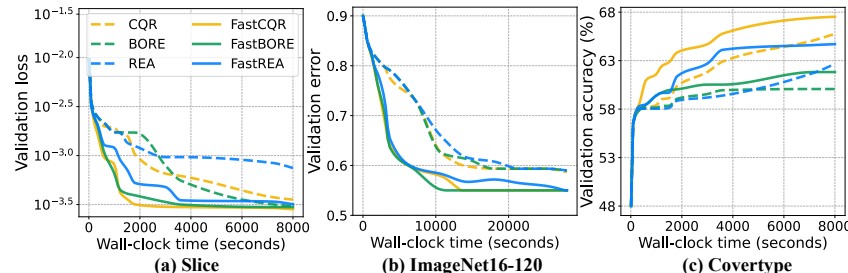

**(a) Slice**   **(b) ImageNet16-120**   **(c) Covertype**

Figure 4: Performance of single-fidelity methods CQR, BORE, REA and their multi-fidelity variants using our extension method.

to as FastCQR, FastBORE, and FastREA respectively. Similar to FastBO, all the multi-fidelity extensions evaluate the configurations to the adaptively identified efficient point and use the corresponding performances for the subsequent operations. The results on three representative datasets are illustrated in Figure 4 and similar results on other datasets are provided in Appendix A.4.3. We can clearly observe that the multi-fidelity variants with our extension method always outperform their single-fidelity counterparts. It is worth noting that REA is an evolutionary algorithm-based HPO method and is also significantly improved by our extension. The observation highlights the ability of the proposed adaptive strategy to extend any single-fidelity method to the multi-fidelity setting. It also suggests future opportunities to extend other advanced single-fidelity techniques into the multi-fidelity setting.

## 6 CONCLUSION

In this paper, we propose a new model-based multi-fidelity HPO method named FastBO, which adaptively identifies the appropriate fidelity for each configuration to fit the surrogate model and offers high-quality performance while ensuring efficient resource utilization. The advantages are achieved through our introduced concepts of efficient point and saturation point, the proposed techniques of learning curve modeling, and well-designed warm-up and post-processing stages with judicious early-termination detection and efficient saturation-level evaluation. Moreover, the proposed adaptive fidelity identification strategy provides a simple way to extend any single-fidelity method to the multi-fidelity setting. Our empirical evaluation demonstrates the effectiveness and wide generality of our proposed techniques. FastBO source code to reproduce our results is freely available at [url omitted].

## REPRODUCIBILITY STATEMENT

The detailed hyperparameter setting of FastBO is provided in § 5, and the detailed hyperparameter setting of the baseline methods can be found in Appendix A.5.2. We use publicly available tabular benchmarks that allow running the experiments without large computational costs. Detailed information on the benchmarks can be found in Appendix A.5.1. In addition, we include the code for our method in the supplementary material.

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

# A  APPENDIX

## CONTENTS

Table 3: The notations used throughout the paper and the corresponding definitions.

| Notation | Definition |
|---|---|
| $a$ | Acquisition function. |
| $c_j(r\|\boldsymbol{\theta}_j), \mathcal{C}(r\|\boldsymbol{\phi})$ | One of, and the combined parametric learning curve model. |
| $\mathcal{C}_i(r)$ | Empirical learning curve for $\boldsymbol{\lambda}_i$. |
| $\mathcal{D}_i$ | Observation set that used to fit the surrogate model, containing $i$ pairs of data points. |
| $e_i$ | The efficient point of $\boldsymbol{\lambda}_i$. |
| $f(\boldsymbol{\lambda}), f(\boldsymbol{\lambda}, r)$ | Performance with configuration $\boldsymbol{\lambda}$ in the single-fidelity and multi-fidelity settings. |
| $k$ | The number of configurations to be promoted. |
| $\mathcal{M}$ | Surrogate model. |
| $\mathcal{O}_i^w$ | Early observation set of $\boldsymbol{\lambda}_i$ across different fidelities, with a maximum level $w$ considered. |
| $r, r_{max}, r_{min}$ | Fidelity; the maximum and minimum fidelity. |
| $s_i$ | The saturation point of $\boldsymbol{\lambda}_i$. |
| $w$ | Warm-up point for all the configurations. |
| $y_i, y_i^r$ | Evaluation results of $f(\boldsymbol{\lambda}_i)$ and $f(\boldsymbol{\lambda}_i, r)$ in the single-fidelity and multi-fidelity settings. |
| $\alpha$ | Performance decrease ratio. |
| $\delta_1, \delta_2$ | Small thresholds used in identifying efficient points and saturation points. |
| $\boldsymbol{\theta}_j, \boldsymbol{\phi}$ | Parameters in one of, and the combined parametric learning curve model. |
| $\lambda_i, \boldsymbol{\lambda}$ | A hyperparameter and a hyperparameter configuration. |
| $\Lambda_i, \boldsymbol{\Lambda}$ | Domain of $\lambda_i$ and search space of $\boldsymbol{\lambda}$. |
| $\omega_j$ | The weight of a parametric learning curve model. |

## A.1 Notation

In Table 3, we provide a comprehensive summary of the notations utilized throughout the paper, along with their detailed definitions and explanations.

## A.2 Illustration on Efficient Point and Saturation Point

In § 4.1, we provide formal definitions for the efficient point and saturation point. Here, we provide a more intuitive understanding of the concepts.

Figure 5 shows an intuitive visualization of two learning curves $\mathcal{C}_1(r), \mathcal{C}_2(r)$, together with their respective efficient points $e_1, e_2$ and saturation points $s_1$, $s_2$. We can easily grasp that the saturation points signify that the performance has nearly reached full convergence, while the efficient points, located at a relatively earlier stage, represent a position where performance can be achieved with high efficiency.

From Figure 5, we can clearly see a significant difference in the shapes of the two learning curves. $\mathcal{C}_2(r)$ experiences rapid initial descent and quick convergence; while $\mathcal{C}_1(r)$ experiences a slower initial descent, but eventually converges to a better performance than $\mathcal{C}_2(r)$. Due to this difference, we can find a crossing point where the two curves meet. Suppose that $\mathcal{C}_1(r)$ and $\mathcal{C}_2(r)$ correspond to configurations $\boldsymbol{\lambda}_1$ and $\boldsymbol{\lambda}_2$ respectively, we can know $\boldsymbol{\lambda}_1$ outranks $\boldsymbol{\lambda}_2$ in terms of configuration performance ranking. Since FastBO

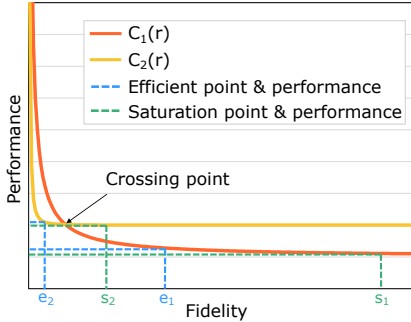

Figure 5: Illustration of efficient point and saturation point associated with learning curves.

utilizes efficient points $e_1$ and $e_2$ as their fidelities for fitting the surrogate model, it is able to capture the distinctive trends in the learning curves of $\boldsymbol{\lambda}_1$ and $\boldsymbol{\lambda}_2$. This ensures that the observed performance $y_1^{e_1}$ surpasses $y_2^{e_2}$, i.e., consistent with the configuration performance ranking, where both $y_1^{e_1}$ and $y_2^{e_2}$ are used to update the surrogate model. In contrast, existing successive halving-based methods may fail to maintain ranking consistency. Specifically, they are susceptible to erroneous termination of $\boldsymbol{\lambda}_1$ if the decision is made before the crossing point. Even with the aid of surrogate models, fitting before the crossing point leads to an inaccurate surrogate model.

Furthermore, we can observe that there is often a gap between the saturation point and the final fidelity, which becomes more pronounced on curves that converge rapidly, such as $\mathcal{C}_2$. FastBO utilizes the saturation point as an approximation

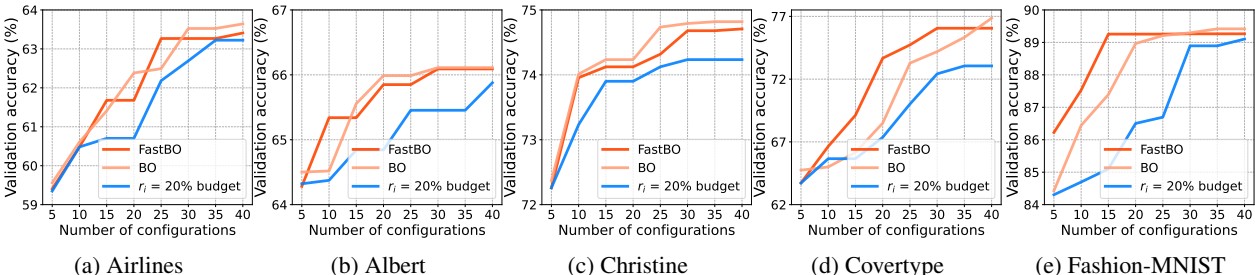

|     |     |     |     |     |
| --- | --- | --- | --- | --- |
| (a) Airlines | (b) Albert | (c) Christine | (d) Covertype | (e) Fashion-MNIST |

Figure 6: Performance of average validation accuracy against the number of evaluated configurations on LCBench.

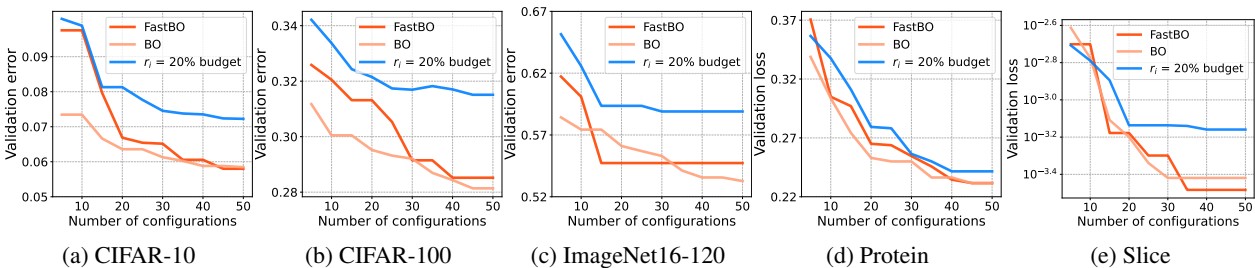

|     |     |     |     |     |
| --- | --- | --- | --- | --- |
| (a) CIFAR-10 | (b) CIFAR-100 | (c) ImageNet16-120 | (d) Protein | (e) Slice |

Figure 7: Performance of average validation error against the number of evaluated configurations on the NAS-Bench-201 benchmark (i.e. (a)-(c)) and performance of average validation loss against the number of evaluated configurations on the FCNet benchmark (i.e., (d) and (e)).

for the final fidelity. Intuitively, it can achieve performance that is very close to the performance at the final fidelity while saving a considerable amount of computational cost.

### A.3 EXPERIMENTS OF SAMPLE EFFICIENCY

In § 5.1, we compare the anytime performance of FastBO with a wide range of HPO methods. One reason for FastBO's good anytime performance can be attributed to its good sample efficiency. Sample efficiency refers to the ability of an algorithm to find the optimal solution with the minimum number of samples. In the context of HPO, sample efficiency quantifies how effectively the algorithm explores the hyperparameter space and identifies promising configurations while minimizing the number of evaluated configurations. Methods with higher sample efficiency, such as BO, are capable of identifying satisfactory configurations with fewer evaluations.

In order to investigate the sample efficiency of FastBO, we conduct experiments using the same settings as the experiments in § 5.1 but plotting the achieved performance as a function of the number of evaluated configurations. Figures 6 and 7 illustrate the results obtained on the three benchmarks. We can see that FastBO is able to achieve comparable, and in some cases, even superior performance to vanilla BO. It is particularly noteworthy considering that FastBO only performs partial evaluations of the configurations and is unsure about their performance at the final fidelity. The results demonstrate that FastBO has the ability to identify the appropriate fidelity for each configuration that can reliably indicate its performance. This remarkable ability is achieved by our proposed adaptive strategy that adaptively finds the efficient point $e_i$ for each configuration $r_i$ as its fidelity $r_i$ for surrogate model fitting.

In order to facilitate a clearer comparison, we also incorporate the results on an additional baseline: a partial evaluation scheme that replaces the adaptive strategy with the adoption of a fixed value as the fidelity for all the configurations to fit the surrogate model. Specifically, we set the fixed fidelity to 20% of the total resource budget and present the corresponding results in Figures 6 and 7. It can be observed that this partial evaluation baseline consistently lags behind both FastBO and vanilla BO. It underscores the challenge of using a fixed fidelity value for all configurations in reflecting their final fidelity performance, which highlights the importance of the adoption of our adaptive strategy.

Table 4: Comparison of relative efficiency for configuration identification. Wall-clock time (abbr. WC time) reports the elapsed time spent for each method on finding configurations with similar performance metrics, i.e., validation error ($\times 10^{-2}$) and validation loss ($\times 10^{-2}$). Regarding relative efficiency, FastBO is set as the baseline with a relative efficiency of 1.00.

| Metric \ Method / Dataset | **FastBO** | BO | PASHA | A-BOHB | A-CQR | BOHB | Hyper-Tune |
|---|---|---|---|---|---|---|---|
| **Airlines** Val. error | $\mathbf{36.2_{\pm 0.1}}$ | $36.3_{\pm 0.5}$ | $\mathbf{36.2_{\pm 0.1}}$ | $36.3_{\pm 0.3}$ | $38.9_{\pm 0.5}$ | $38.5_{\pm 0.1}$ | $\mathbf{36.2_{\pm 0.1}}$ |
| WC time (h) | $\mathbf{0.5_{\pm 0.3}}$ | $2.4_{\pm 1.3}$ | $1.1_{\pm 0.7}$ | $1.1_{\pm 0.6}$ | $2.7_{\pm 0.6}$ | $2.2_{\pm 0.4}$ | $1.1_{\pm 0.6}$ |
| Rel. efficiency | **1.00** | 0.23 | 0.51 | 0.36 | 0.20 | 0.25 | 0.48 |
| **Albert** Val. error | $\mathbf{33.9_{\pm 0.1}}$ | $34.0_{\pm 0.1}$ | $34.3_{\pm 0.1}$ | $34.0_{\pm 0.0}$ | $34.8_{\pm 0.7}$ | $34.7_{\pm 0.2}$ | $34.0_{\pm 0.3}$ |
| WC time (h) | $\mathbf{0.5_{\pm 0.3}}$ | $1.0_{\pm 0.7}$ | $1.2_{\pm 0.8}$ | $1.6_{\pm 1.0}$ | $3.2_{\pm 0.4}$ | $1.9_{\pm 1.4}$ | $1.2_{\pm 1.1}$ |
| Rel. efficiency | **1.00** | 0.48 | 0.39 | 0.28 | 0.14 | 0.24 | 0.39 |
| **Christine** Val. error | $\mathbf{25.3_{\pm 0.1}}$ | $25.5_{\pm 0.1}$ | $25.6_{\pm 0.1}$ | $25.5_{\pm 0.1}$ | $26.7_{\pm 0.0}$ | $26.8_{\pm 0.2}$ | $25.4_{\pm 0.0}$ |
| WC time (h) | $\mathbf{0.8_{\pm 0.3}}$ | $2.4_{\pm 1.3}$ | $2.4_{\pm 2.2}$ | $2.1_{\pm 1.2}$ | $1.6_{\pm 2.1}$ | $1.5_{\pm 0.9}$ | $2.9_{\pm 0.8}$ |
| Rel. efficiency | **1.00** | 0.33 | 0.33 | 0.37 | 0.48 | 0.54 | 0.27 |
| **Fashion-MNIST** Val. error | $\mathbf{10.7_{\pm 0.1}}$ | $\mathbf{10.7_{\pm 0.1}}$ | $\mathbf{10.7_{\pm 0.1}}$ | $\mathbf{10.7_{\pm 0.1}}$ | $11.6_{\pm 0.3}$ | $11.4_{\pm 0.2}$ | $\mathbf{10.7_{\pm 0.1}}$ |
| WC time (h) | $\mathbf{0.2_{\pm 0.1}}$ | $0.8_{\pm 0.7}$ | $1.8_{\pm 1.4}$ | $0.5_{\pm 0.2}$ | $2.5_{\pm 1.1}$ | $3.2_{\pm 0.8}$ | $0.6_{\pm 0.4}$ |
| Rel. efficiency | **1.00** | 0.21 | 0.10 | 0.34 | 0.07 | 0.19 | 0.27 |
| **CIFAR-10** Val. error | $\mathbf{6.2_{\pm 0.4}}$ | $6.5_{\pm 0.4}$ | $6.4_{\pm 0.7}$ | $\mathbf{6.2_{\pm 0.2}}$ | $6.3_{\pm 0.4}$ | $6.3_{\pm 0.2}$ | $\mathbf{6.2_{\pm 0.2}}$ |
| WC time (h) | $\mathbf{0.6_{\pm 0.4}}$ | $3.9_{\pm 2.0}$ | $1.3_{\pm 0.6}$ | $2.3_{\pm 1.1}$ | $2.6_{\pm 0.9}$ | $2.1_{\pm 0.5}$ | $1.6_{\pm 0.8}$ |
| Rel. efficiency | **1.00** | 0.16 | 0.49 | 0.27 | 0.25 | 0.31 | 0.39 |
| **CIFAR-100** Val. error | $\mathbf{28.7_{\pm 1.3}}$ | $29.6_{\pm 1.4}$ | $32.8_{\pm 8.9}$ | $\mathbf{28.7_{\pm 1.2}}$ | $28.8_{\pm 1.5}$ | $28.8_{\pm 0.7}$ | $29.4_{\pm 1.1}$ |
| WC time (h) | $\mathbf{1.2_{\pm 0.9}}$ | $2.4_{\pm 1.6}$ | $1.6_{\pm 1.4}$ | $2.8_{\pm 1.2}$ | $2.8_{\pm 1.3}$ | $1.7_{\pm 0.4}$ | $1.7_{\pm 0.5}$ |
| Rel. efficiency | **1.00** | 0.50 | 0.73 | 0.43 | 0.42 | 0.72 | 0.72 |
| **Protein** Val. loss | $\mathbf{22.6_{\pm 0.4}}$ | $22.9_{\pm 0.7}$ | $23.6_{\pm 0.9}$ | $22.6_{\pm 0.3}$ | $22.7_{\pm 0.5}$ | $23.2_{\pm 0.4}$ | $22.7_{\pm 0.7}$ |
| WC time (h) | $\mathbf{0.3_{\pm 0.1}}$ | $1.2_{\pm 0.7}$ | $0.7_{\pm 0.6}$ | $0.8_{\pm 0.5}$ | $0.6_{\pm 0.3}$ | $1.3_{\pm 0.7}$ | $1.1_{\pm 0.5}$ |
| Rel. efficiency | **1.00** | 0.23 | 0.38 | 0.32 | 0.42 | 0.21 | 0.25 |

## A.4 EXTENDED EXPERIMENTS

In this section, we provide additional experimental results running on more datasets, including the experiments to investigate the efficiency in identifying high-quality configurations, the effectiveness of the proposed adaptive fidelity identification strategy, and the generality of the extension method.

### A.4.1 EXTENDED EXPERIMENTS OF EFFICIENCY ON CONFIGURATION IDENTIFICATION

In § 5.2, we compare the time spent for the HPO methods on identifying a satisfactory configuration. Here, we report additional results on the datasets from the LCBench, NAS-Bench-201 and FCNet benchmarks in Table 4. For the "Airlines", "Albert", "Christine" and "Fashion-MNIST" datasets from LCBench, we convert their performance metric from validation accuracy to validation error for better readability.

The experimental results shown in Table 4 are consistent with those shown in § 5.2. For the "rel. efficiency" rows, FastBO is set as the baseline with a relative efficiency of 1.00, and then we compute the relative efficiency of other methods. We can observe that FastBO saves considerable wall-clock time over the baseline methods when achieving similar or better performance values, demonstrating the high efficiency of FastBO in identifying a good configuration. The model-free PASHA method often gets a high variance in wall-clock time because different random seeds can have a larger impact on it. Results of other model-free methods are not included in Table 4, since PASHA demonstrates its superiority over them (Bohdal et al., 2023).

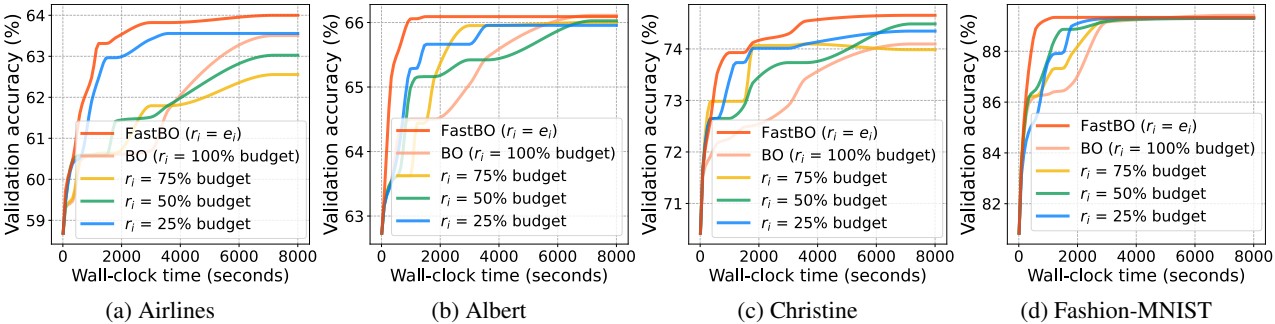

|           |           |           |           |
|-----------|-----------|-----------|-----------|
| (a) Airlines | (b) Albert | (c) Christine | (d) Fashion-MNIST |

Figure 8: Average validation accuracy on the LCBench benchmark of (i) FastBO that set $r_i = e_i$, (ii) the schemes that use fixed 25%, 50%, 75% of the total resource budget as $r_i$ for all configurations, and (iii) vanilla BO that uses 100% total resource budget as $r_i$.

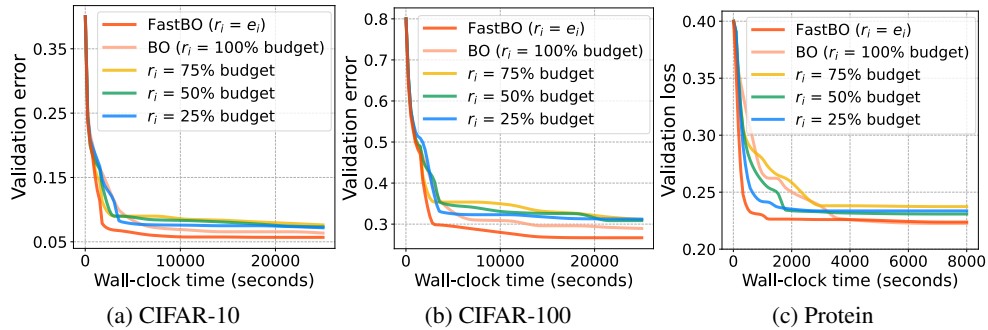

|           |           |           |
|-----------|-----------|-----------|
| (a) CIFAR-10 | (b) CIFAR-100 | (c) Protein |

Figure 9: Average validation error on the NAS-Bench-201 benchmark (including the "CIFAR-10", "CIFAR-100" datasets) and average validation loss on the FCNet benchmark (including the "Protein" dataset) of (i) FastBO that set $r_i = e_i$, (ii) the schemes that use fixed 25%, 50%, 75% of the total resource budget as $r_i$ for all configurations, and (iii) vanilla BO that uses 100% total resource budget as $r_i$.

### A.4.2 EXTENDED EXPERIMENTS OF EFFECTIVENESS OF EFFICIENT POINT

In § 5.3, we examine the effectiveness of the proposed adaptive fidelity identification strategy. Here, we provide additional results on more datasets.

We show the results on the LCBench benchmark in Figure 8 and the results on the NAS-Bench-201 and FCNet benchmarks in Figure 9. Specifically, our FastBO with the adaptive fidelity identification strategy sets the efficient point $e_i$ for each configuration $\lambda_i$ as its fidelity $r_i$ to fit the surrogate model. In contrast, the vanilla BO is a full evaluation scheme that uses 100% of the total resource budget as $r_i$. The other three baselines are also partial evaluation schemes like FastBO but without the adaptive fidelity identification strategy. They replace the adaptive choice of $r_i = e_i$ with a fixed fidelity, including 25%, 50%, and 75% of the total resource budget, for all the configurations to fit the surrogate model.

The results shown in Figures 8 and 9 are consistent with those shown in § 5.3. We have two main observations. Firstly, FastBO outperforms the other partial evaluation schemes that remove the adaptive fidelity identification strategy, showing the effectiveness of the proposed adaptive strategy. Secondly, although the partial evaluation schemes with fixed $r_i$ are able to converge faster than the full evaluation counterpart (i.e., the vanilla BO) in the initial stage, this early advantage diminishes progressively over time. Finally, these partial evaluation baselines show significant differences in their final performance on 4 out of 7 datasets when compared to vanilla BO. The main reason is that these partial evaluation schemes naively use a fixed $r_i$ for all the configurations and thus fail to create an accurate surrogate model to identify more promising configurations. This observation also highlights the importance of the adoption of our adaptive fidelity identification strategy.

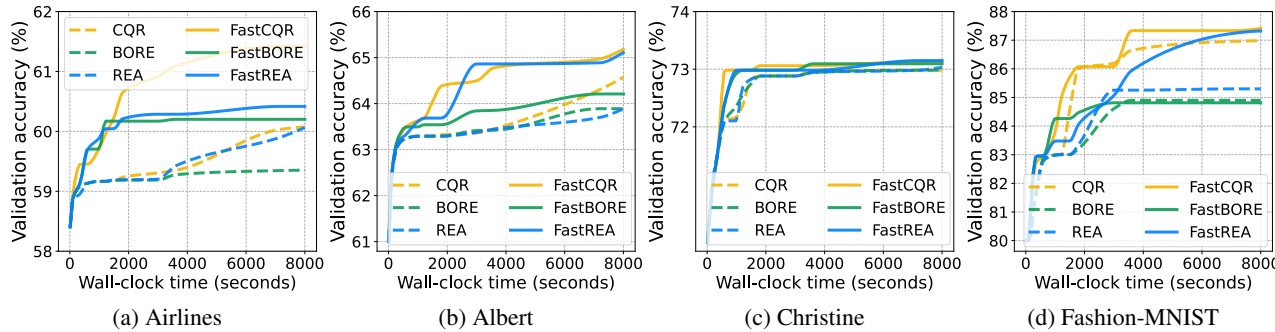

|                |                |                |                |
|:--------------:|:--------------:|:--------------:|:--------------:|
| (a) Airlines   | (b) Albert     | (c) Christine  | (d) Fashion-MNIST |

Figure 10: Performance of single-fidelity methods CQR, BORE, REA and their multi-fidelity variants FastCQR, FastBORE, FastREA using our extension method: average validation accuracy on the LCBench benchmark.

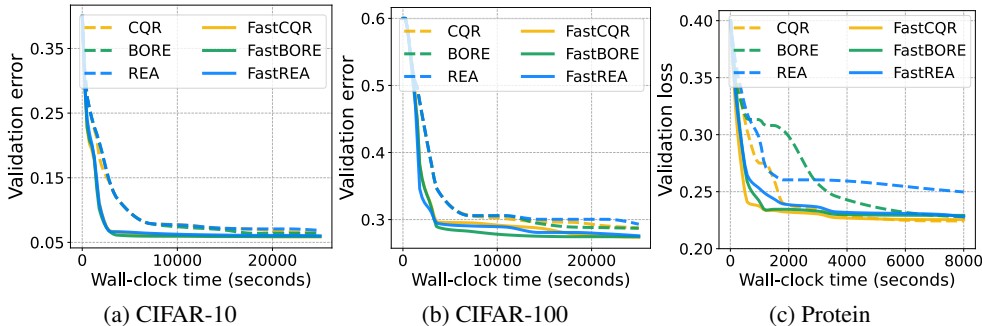

|                |                |                |
|:--------------:|:--------------:|:--------------:|
| (a) CIFAR-10   | (b) CIFAR-100  | (c) Protein    |

Figure 11: Performance of single-fidelity methods CQR, BORE, REA and their multi-fidelity variants FastCQR, FastBORE, FastREA using our extension method: average validation error on NAS-Bench-201 and average validation loss on FCNet.

### A.4.3 EXTENDED EXPERIMENTS OF GENERALITY OF EXTENSION METHOD

In § 5.4, we investigate the ability of our proposed extension method. Here, we provide additional results on the datasets from LCBench in Figure 10 and more results on the datasets from NAS-Bench-201 and FCNet in Figure 11. We run three well-known single-fidelity methods CQR (Salinas et al., 2023), BORE (Tiao et al., 2021), and REA (Real et al., 2019), and extend them to the multi-fidelity setting using our extension method, denoted as FastCQR, FastBORE, and FastREA respectively. More specifically, all the multi-fidelity variants evaluate the configurations to their efficient points and use the corresponding performances for the subsequent operations, i.e., fitting the surrogate model for FastCQR and FastBORE, selection and variation for FastREA.

From Figures 10 and 11, we can clearly observe that the multi-fidelity variants with our extension method always outperform their single-fidelity counterparts. For the relatively simple task presented by the "Christine" dataset, the distinctions are not as pronounced as they are in the case of other datasets. However, it is still evident that the multi-fidelity methods are able to converge towards a higher accuracy more rapidly. Moreover, the evolutionary algorithm REA can also be enhanced by our extension method. The results are consistent with the observations shown in § 5.4 and highlight the wide applicability of the proposed adaptive strategy to extend any single-fidelity method to the multi-fidelity setting.

### A.5 EXPERIMENTAL SETUP

Here we provide more details on the experimental setup, including details of the used benchmarks and choice of parameters on the baseline methods.

Table 5: Detailed information of LCBench, NAS-Bench-201 and FCNet benchmarks.

| Benchmark | #Evaluations | #Hyperparameters | #Fidelities |
|---|---|---|---|
| LCBench | 2,000 | 7 | 50 |
| NAS-Bench-201 | 15,625 | 6 | 200 |
| FCNet | 62,208 | 9 | 100 |

### A.5.1 BENCHMARK DETAILS

In our experiments, we use 3 well-known tabular benchmarks: LCBench (Zimmer et al., 2021), NAS-Bench-201 (Dong & Yang, 2020), and FCNet (Klein & Hutter, 2019). We conclude detailed information on these benchmarks in Tables 5, including the number of provided evaluations, the number of hyperparameters, and the number of fidelities. Table 6 provides information on the hyperparameters in the benchmarks and their corresponding configuration spaces.

**LCBench.** LCBench is a neural network benchmark that consists of 2000 hyperparameter configurations. LCBench features a search space of 7 numerical hyperparameters of neural networks, including the number of layers, the maximum number of units per layer, batch size, learning rate, weight decay, momentum, and dropout. The fidelity refers to the number of epochs in LCBench and each hyperparameter configuration is trained for 50 epochs. LCBench contains 35 datasets and we run the 5 most expensive ones.

**NAS-Bench-201.** NAS-Bench-201 is a benchmark that consists of 15625 hyperparameter configurations. NAS-Bench-201 features a search space of 6 categorical hyperparameters that correspond to 6 operations within the macro architecture cell. The fidelity refers to the number of epochs in NAS-Bench-201 and each hyperparameter configuration, which represents a network architecture, is trained for 200 epochs. NAS-Bench-201 contains the image classification datasets cifar-10, cifar-100 and ImageNet16-120.

**FCNet.** FCNet is a benchmark that consists of 62208 hyperparameter configurations. FCNet features a search space of 4 architectural choices (i.e., the number of units and activation functions for two layers) and 5 hyperparameters (i.e., dropout rates per layer, batch size, initial learning rate and learning rate schedule). The fidelity refers to the number of epochs in FCNet and each hyperparameter configuration is trained for 100 epochs. FCNet uses 4 popular UCI datasets for regression.

### A.5.2 CHOICE OF PARAMETERS ON BASELINE METHODS

We use implementations of all the baseline HPO methods provided in Syne Tune (Salinas et al., 2022). We here list the parameters used for running the baselines in our experiments. In general, we follow the default settings in Syne Tune which are also recommended in the previous work.

- Vanilla Bayesian Optimization (BO) (Snoek et al., 2012) uses a Matérn $\frac{5}{2}$ kernel with automatic relevance determination parameters and the expected improvement (EI) acquisition function.
- ASHA (Li et al., 2020), Hyperband (Li et al., 2017) and PASHA (Bohdal et al., 2023) follow the successive halving (SHA) (Jamieson & Talwalkar, 2016) framework and sample new hyperparameter configurations at random. We use the recommended reduction factor $\eta$ of 3 in all of them. In other words, the evaluations are stopped after 1, 3, 9, 27, ... resource levels.
- A-BOHB (Klein et al., 2020) follows the SHA framework with $\eta = 3$. It uses a stopping variant asynchronous scheduling, which is different from the promotion variant asynchronous scheduling used in ASHA. New configurations are selected as in the vanilla BO.
- A-CQR (Salinas et al., 2023) follows the SHA framework with $\eta = 3$ and uses the promotion variant asynchronous scheduling as ASHA. It uses BO to select the configuration and uses the last observed values from the SHA framework to fit the surrogate model. It uses a conformal quantile regression-based surrogate model.
- BOHB (Falkner et al., 2018) follows the SHA framework with $\eta = 3$ and uses synchronous scheduling. It uses BO with a multi-variate kernel density estimator to select new configurations.
- Hyper-Tune (Li et al., 2022) follows the SHA framework with $\eta = 3$ and uses the promotion variant asynchronous scheduling as ASHA. It fits independent Gaussian process models at different resource levels.

Table 6: Hyperparameters and configuration spaces for benchmarks.

| Benchmark | Hyperparameter | Configuration space |
|---|---|---|
| LCBench | num_layers | [1, 5] |
| | max_units | [64, 512] |
| | batch_size | [16, 512] |
| | learning_rate | [1e-4, 1e-1] |
| | weight_decay | [1e-5, 0.1] |
| | momentum | [0.1, 0.99] |
| | max_dropout | [0.0, 1.0] |
| NAS-Bench-201 | x0 | [avg_pool_3x3, nor_conv_3x3, skip_connect, nor_conv_1x1, none] |
| | x1 | [avg_pool_3x3, nor_conv_3x3, skip_connect, nor_conv_1x1, none] |
| | x2 | [avg_pool_3x3, nor_conv_3x3, skip_connect, nor_conv_1x1, none] |
| | x4 | [avg_pool_3x3, nor_conv_3x3, skip_connect, nor_conv_1x1, none] |
| | x3 | [avg_pool_3x3, nor_conv_3x3, skip_connect, nor_conv_1x1, none] |
| | x5 | [avg_pool_3x3, nor_conv_3x3, skip_connect, nor_conv_1x1, none] |
| FCNet | activation_1 | [tanh, relu] |
| | activation_2 | [tanh, relu] |
| | batch_size | [8, 16, 32, 64] |
| | dropout_1 | [0.0, 0.3, 0.6] |
| | dropout_2 | [0.0, 0.3, 0.6] |
| | init_lr | [0.0005, 0.001, 0.005, 0.01, 0.05, 0.1] |
| | lr_schedule | [cosine, const] |
| | n_units_1 | [16, 32, 64, 128, 256, 512] |
| | n_units_2 | [16, 32, 64, 128, 256, 512] |

The experiments in § 5.4 contain three HPO methods and we use implementations of them provided in Syne Tune. We also provide the parameter settings of the three methods as follows.

- CQR (Salinas et al., 2023) uses BO with a conformal quantile regression-based surrogate model to select new configurations.

- BORE (Tiao et al., 2021) is evaluated with XGBoost (Chen & Guestrin, 2016) as the classifier with its default setting. We set $\gamma = 1/4$, consistent with BORE's default hyperparameter setting.

- REA (Real et al., 2019) is an evolutionary algorithm that uses a population size of 10, and 5 samples are drawn to select a mutation from.

