# OpenReview forum: "Efficient Hyperparameter Optimization with Adaptive Fidelity Identification"
_ICLR.cc/2024/Conference — ICLR 2024 Conference Desk Rejected Submission_

### Official Review · Reviewer_WVYX · 2023-10-23

**Soundness:** 2 fair
**Presentation:** 3 good
**Contribution:** 2 fair
**Rating:** 3
**Confidence:** 5

**Summary:**

The authors propose a multi-fidelity BO method for HPO optimization with adaptive fidelity identification. The authors use a GP as a surrogate model combined with EI as an acquisition function. The authors use a combination of learning curve models to estimate the fidelity budget that accurately represents the performance of the hyperparameter configuration (efficient point) based on the partial observations obtained from a warmup phase. In the end, the top $k$ configurations are trained until their saturation points.

The authors provide results on 10 datasets out of 3 diverse benchmarks (LCBench, NB201, FCNet). The authors additionally apply their algorithm components in other non-multi-fidelity surrogate models and present the improvement in results.

**Strengths:**

- The work has a good structure and is well-written.
- I found the application of the proposed algorithm on methods that do not support multi-fidelity optimization particularly interesting.

**Weaknesses:**

- The coverage of related work is lacking, in particular, the authors have missed two related multi-fidelity model-based baselines that feature a dynamic schedule:


    **DyHPO** [1], a deep GP kernel multi-fidelity method that does not follow a SHA schedule, but a dynamic one, moving one step at a time and predicting the performance for the next step of different hyperparameter configurations.

    **DPL** [2], a deep ensemble that conditions the predictions to follow a power law formulation, also following a dynamic schedule, moving one step at a time, however, in contrast to DyHPO it predicts the final validation performance for every configuration and pushes the predicted best configuration with one step.

- The paper advocates that it is able to adaptively decide the fidelity for which every configuration should be evaluated, however, it incorporates fixed design choices, e.g $w$, the number of warmup steps necessary to have for every learning curve is 20% of the total curve (Although the authors have a heuristic that can stop the warmup phase if no improvement is observed).
- The combination of a surrogate model together with LC prediction for an individual hyperparameter configuration can be inefficient from my perspective.

[1] Wistuba et al. "Supervising the multi-fidelity race of hyperparameter configurations." Advances in Neural Information Processing Systems 35 (2022): 13470-13484.

[2] Kadra et al. "Power Laws for Hyperparameter Optimization." Thirty-seventh Conference on Neural Information Processing Systems (2023)

**Questions:**

- **In related work:** PAHSA -> PASHA
- **In related work (Multi-Fidelity setting):** The best performance does not necessarily need to be at $r_{max}$ for the incumbent configuration, it could also be at an intermediate fidelity level. The proposed target of optimization is not correct and the rest of the paper is based on that.  As the authors note later on: "However, we observe that an increase in fidelities does not always result in performance improvement."
- **In section 4.2:** $w$ is used, although it is defined only in 4.3.
- Unless I missed it, the surrogate model is referred to as $M$, however, no information is given on what the surrogate model is, until Section 5 with **FastBO uses a Matern $\frac{5}{2}$ kernel**. I believe it should be clearly described in the manuscript that $M$ is a GP.
- **"It is worth noting that the additional time required is factored into the overall time."** -> Is the benchmark configuration evaluation time, and the warmup stage included in the wall-clock time? Is the wall-clock time multiplied by 4 since there are 4 parallel workers?
- Why are there only 10 datasets selected from LCBench, FCNET, and NB201? What was the criterion used for the selection?
LCBench alone has 35 datasets.
- Could the authors provide a summary plot of the ranks that all the methods achieve in the different benchmarks?
- Could the authors elaborate on whether the difference in results is statistically significant?

---

> ### Author Response · Authors · 2023-11-28
> **Response posted by the chairs on behalf of the authors (no need to respond)**
>
> To Reviewer WVYX:
>
> We appreciate the effort and time spent reviewing our submission, and we are grateful that you highlighted the wide generality of our method. We address the weaknesses and questions next.
>
> > “The coverage of related work is lacking, …, DyHPO, …, DPL, …”
>
> We sincerely thank you for bringing DPL to our attention, which is a latest work that had not been accepted at the time of our paper drafting, limiting our awareness of its content. We have added additional experiments and discussions on DyHPO. It performs well on both LCBench and NAS-Bench-201, but exhibits slightly inferior performance on FCNet. We found these two works interesting and appreciate their contributions to the community. These important works have also been incorporated into the related work.
>
> In contrast to the two methods that introduce new surrogates and use neural networks as support, our FastBO takes a different way. We opt for a more explicit way to incorporate learning curve information. Instead of using black-box neural networks, our method provides enhanced interpretability and ease of understanding. This is possible because the concepts of efficient and saturation points are transparent. Furthermore, our strategy can be easily extended to any single-fidelity method without the need for a redesign.
>
>
> > “The paper advocates that it is able to adaptively decide the fidelity for which every configuration should be evaluated, however, it incorporates fixed design choices, e.g $w$, the number of warmup steps necessary to have for every learning curve is 20% of the total curve (Although the authors have a heuristic that can stop the warmup phase if no improvement is observed).”
>
> Because of the adaptive process of efficient and saturation points customization, our method has the capability to adaptively decide the fidelity for each configuration. This adaptive process is different from the warm-up process, where $w$ is used.
>
> Regarding the concern on the fixed design choices of $w$, we have added experiments to study the impact of different choices of $w$. The results show that FastBO is not highly sensitive to the values of $w$, showing the robustness of our method. The fixed choice of $w$ does not compromise our key contributions. We intentionally kept a fixed setting for $w$ to avoid introducing unnecessary complexity by tuning a 'hyperparameter' within a 'hyperparameter' optimization algorithm. Fine-tuning $w$ is a possibility and can lead to a further improvement on performance.
>
>
> > “The combination of a surrogate model together with LC prediction for an individual hyperparameter configuration can be inefficient from my perspective.”
>
> FastBO is in fact efficient. We added a comparative analysis between FastBO and DyHPO, a representative method that integrates LC prediction into the surrogate model. The results show that the overhead of FastBO is smaller than DyHPO.
>
>
> > “The best performance does not necessarily need to be at $r_{max}$ for the incumbent configuration, it could also be at an intermediate fidelity level. The proposed target of optimization is not correct and the rest of the paper is based on that. As the authors note later on: "However, we observe that an increase in fidelities does not always result in performance improvement."”
>
> We acknowledge the confusion caused by the statement in Section 3 and have corrected it. However, the rest of the paper remains unaffected. In Section 4.3, where we state, "However, we observe that an increase in fidelities does not always result in performance improvement," this claim is correct and does not conflict with the multi-fidelity target. The purpose is just to present an implementation detail in post-processing: recording all intermediate results and selecting the final output from the intermediates. While it is common in implementation, we included it for the completeness of our method. We have rephrased Section 4.3.
>
>
> > “In section 4.2: $w$ is used, although it is defined only in 4.3.”
>
> Thanks for your suggestion. We have updated Section 4.2 to avoid any confusion.
>
>
> > “Unless I missed it, the surrogate model is referred to as $M$, however, no information is given on what the surrogate model is, until Section 5 with FastBO uses a Matern $\frac{5}{2}$ kernel. I believe it should be clearly described in the manuscript that $M$ is a GP.”
>
> Thanks for your suggestion. We have explicitly mentioned the used surrogate model as well as the acquisition function in Section 3. Our focus is not to introduce new surrogate models or optimize existing ones, and our method can be applied to any surrogate model. Therefore, we did not highlight this aspect earlier.

---

> > ### Author Response · Authors · 2023-11-28
> > **Response (part 2) posted by the chairs on behalf of the authors (no need to respond)**
> >
> > > “Is the benchmark configuration evaluation time, and the warmup stage included in the wall-clock time? Is the wall-clock time multiplied by 4 since there are 4 parallel workers?”
> >
> > For the first question, yes, all the time you mentioned are included in the wall-clock time. For the second question, no, the wall-clock time is not multiplied by 4 because the workers are in parallel. The wall-clock time accounts for the time spent on evaluations (provided by the tabular benchmark), HPO methods, scheduling, etc. We use a 4-worker setting in our experiments, the same as the experimental setting in PASHA[1] and CQR[2].
> >
> >
> > > “Why are there only 10 datasets selected from LCBench, FCNET, and NB201? What was the criterion used for the selection? LCBench alone has 35 datasets.”
> >
> > For LCBench, we selected the 5 most expensive datasets among the 35 datasets available, consistent with the setting in CQR [2]. The same selection criterion was used for FCNet. Note that the 3 benchmarks used are different in their maximum fidelity levels and performance metrics, which we believe is enough to prove the effectiveness of our method.
> >
> >
> > > “Could the authors provide a summary plot of the ranks that all the methods achieve in the different benchmarks?”
> >
> > > “Could the authors elaborate on whether the difference in results is statistically significant?”
> >
> > Thanks for your suggestion! We have added a separate section to include critical difference diagram to summarize the ranks of all methods and provide information on the statistical difference on each benchmark. The difference in results
> >
> >
> > [1] Bohdal et al. "PASHA: efficient HPO and NAS with progressive resource allocation". ICLR (2023).
> >
> > [2] Salinas et al. "Optimizing hyperparameters with conformal quantile regression." ICML (2023).

---

### Official Review · Reviewer_ZiMT · 2023-10-27

**Soundness:** 3 good
**Presentation:** 3 good
**Contribution:** 3 good
**Rating:** 6
**Confidence:** 4

**Summary:**

Multi-fidelity optimization represents the current state-of-the-art in hyperparameter optimization. The concept behind it is to terminate the evaluation of poorly performing hyperparameter configurations early in order to allocate more resources to promising configurations. A crucial element of this approach is to model the learning curve, i.e validation performance after each epoch, of a hyperparameter configuration.

The paper introduces a novel multi-fidelity strategy that aims to predict key points of a learning curve. After some initial observations, the proposed method determines the so-called efficiency point, where performance improvements start to flatten out. The evaluation of this hyperparameter configuration is then continued until this efficiency point. Furthermore, for each hyperparameter configuration the saturation point is computed, which is the epoch level where the learning curve is not improving anymore. After optimization, the top-k hyperparameter configurations are evaluated until this saturation point.

**Strengths:**

- Overall, I found the paper to be well-written and easy to follow. Additionally, I think it nicely captures the relevant literature. The proposed method is well-motivated, and the various steps are convincing.

- The proposed method outperforms state-of-the-art methods across a range of benchmarks from the HPO literature. However, I do have some questions regarding these results (see below).

**Weaknesses:**

- The method introduces additional hyperparameters \delta_1, \delta_2. These are not simple to interpret and hence might be difficult to set a-priori in practice. It would be also great if the paper could comment how important it is to set these values correctly.



- The proposed learning curve model appears to be quite similar to that of Domhan et al., with the key difference being that it does not consider the epistemic uncertainty of the individual models. In contrast, Domhan et al. employed a Bayesian approach to obtain meaningful uncertainty estimates for their predictions. While the paper argues that Domhan et al.'s method is primarily designed for the highest fidelity, I don't see a reason why it couldn't also be used to predict the efficient or saturation point.

**Questions:**

- The results seem somehow different to the results reported by Salinas et al. For example A-CQR performs worse than A-BOHB on the LCBench benchmark. Did you follow their experimental setup? If not, what has changed and why?


- Did you change the latex template? It feels like the margins are smaller. If so, please make sure to follow the original template.

- Do you factor in the post-processing time into your runtime plots?


- Figure 2: I am somewhat surprised that FastBO outperforms ASHA-based methods so rapidly. My understanding is that FastBO assesses each configuration for at least 20% of the maximum number of epochs, which corresponds to 20 epochs on the FCNet benchmarks and 40 epochs on the NASBench202 benchmark. The smallest resource level for ASHA-based methods is typically set to 1 epoch, which implies that it should be able to evaluate a considerably greater number of configurations than FastBO, right?

**Details Of Ethics Concerns:**

No ethics concerns.

---

> ### Author Response · Authors · 2023-11-28
> **Response posted by the chairs on behalf of the authors (no need to respond)**
>
> To Reviewer ZiMT:
>
> Thank you for the in-depth review with valuable feedback and advice. We appreciate the significant effort spent reviewing our paper and recognizing the strengths of our method. We address the weaknesses and questions next.
>
> > “The method introduces additional hyperparameters \delta_1, \delta_2. These are not simple to interpret and hence might be difficult to set a-priori in practice. It would be also great if the paper could comment how important it is to set these values correctly.”
>
> We appreciate your comments on the introduced hyperparameters. In order to avoid introducing complexity by tuning a 'hyperparameter' within a 'hyperparameter' optimization algorithm, we intentionally set the hyperparameters in a simple way. The setting of $\delta_1$ and $\delta_2$ is related to the performance metrics. We standardized the metrics to a scale from 0 to 1 before setting them. After that, the values (i.e., 0.001 and 0.0005) are just the common thresholds used for detecting minor performance variations. We encourage the users to directly use our default setting. Fine-tuning them is also a possibility and, if explored, may lead to further optimization on performance.
>
>
> > “The proposed learning curve model appears to be quite similar to that of Domhan et al., … I don't see a reason why it couldn't also be used to predict the efficient or saturation point.”
>
> A crucial difference between FastBO and Domhan et al.’s method is the targets of learning curve modeling. Domhan et al. aim to predict how likely the current configuration's final performance beat the current best values. Thus, they need to obtain uncertainty by a Bayesian approach. In contrast, we aim to adaptively identify fidelities for each configuration, with the learning curve modeling only serving as a supporting tool. Thus, we don't need their complex parametric model (e.g., considering 11 learning curves) or to estimate uncertainty. Therefore, using Domhan et al.'s learning curve modeling module may not be a good choice for us, although it could also be used to predict the efficient or saturation point.
>
> Given the efficiency goal of the HPO method, we simplified the parametric learning curve models to strike a balance between capturing essential learning curve shapes and prioritizing computational efficiency. Opting for complex learning curve models introduces a computational burden during parameter estimation, as the computational complexity of parameter estimation is proportional to the number of parameters in the combined learning curve model. The increase in computational demands increases the time required for each configuration, which runs counter to the objective of designing efficient HPO algorithms.
>
> > “The results seem somehow different to the results reported by Salinas et al. … Did you follow their experimental setup? If not, what has changed and why?”
>
> We appreciate your observation regarding the experimental comparison. We followed the settings of A-CQR, but we mistakenly set up A-CQR runs with different approaches to estimate missing evaluation points on LCBench. This is a benchmark-related issue and may result in certain differences in datasets. Thanks for pointing it out! We have fixed it and found that A-CQR is generally better than A-BOHB. We will fix the corresponding figures and tables. However, we also found that different runs of A-CQR with the same random seeds sometimes led to unstable results. This is a potential reason for the different results.
>
> > “Do you factor in the post-processing time into your runtime plots?”
>
> Yes, all the FastBO processing time are included in the wall-clock time.

---

> > ### Author Response · Authors · 2023-11-28
> > **Response (part 2) posted by the chairs on behalf of the authors (no need to respond)**
> >
> > > “Figure 2: I am somewhat surprised that FastBO outperforms ASHA-based methods so rapidly. My understanding is that FastBO assesses each configuration for at least 20% of the maximum number of epochs, which corresponds to 20 epochs on the FCNet benchmarks and 40 epochs on the NASBench202 benchmark. The smallest resource level for ASHA-based methods is typically set to 1 epoch, which implies that it should be able to evaluate a considerably greater number of configurations than FastBO, right?”
> >
> > Your understanding is essentially correct. FastBO evaluates many configurations for at least 20% of the maximum number of epochs, but there are some configurations being terminated earlier due to the detection of consecutive performance deterioration. Moreover, ASHA-based methods can evaluate a greater number of configurations than FastBO. This can be observed in the very early stage, where ASHA-based methods often outperform FastBO as they can quickly explore numerous configurations with 1 epoch.
> >
> > However, the ability to evaluate more configurations does not always mean better performance. FastBO excels in partial evaluation while maintaining high sample efficiency at the same time (the experiments on sample efficiency can be found in Appendix A.4), leading to its competitive performance over time. FastBO's ability to adaptively identify fidelities for each configuration contributes to its advantage of fitting the surrogate model more effectively. Some ASHA-based methods are model-free, thus having low sample efficiency. Additionally, ASHA-based methods often limit the evaluations to only constrained fidelities for some time, thus struggling to provide relatively high performance in a short time.

---

### Official Review · Reviewer_gzF7 · 2023-11-03

**Soundness:** 2 fair
**Presentation:** 2 fair
**Contribution:** 2 fair
**Rating:** 5
**Confidence:** 3

**Summary:**

The paper proposes hyper-parameter optimization in which different fidelity of observations are incorporated. The proposed method approximates the learning curve by combining three simple parametric models, and adaptively determine the appropriate stopping point for each observation.

**Strengths:**

The basic idea is clear and the proposed method would be simple to implement.

**Weaknesses:**

Overall, the proposed method consists of several simple heuristics whose justification as a general methodology is not fully clear, and for me, it is somewhat difficult to see significant technical contributions in the paper.

The model of the learning curve is too simple. I do not agree with that these three functions have enough flexibility to handle a variety of shapes of learning curves. In particular, learning curves can have non-convex shapes. In this case, it seems almost impossible to obtain accurate future prediction from warm-up small amount of observations.

Most of well-known BO acquisition functions such as EI, UCB, knowledge gradient, predictive entropy search, and max-value entropy search. A lot of multi-fidelity counter-part has been proposed. The authors ignored most of them.

**Questions:**

Appropriate hyper-parameter values for alpha, delta_1, and delta_2 should depend on a given dataset. How can the current setting be justified?

---

> ### Author Response · Authors · 2023-11-28
> **Response posted by the chairs on behalf of the authors (no need to respond)**
>
> To Reviewer gzF7:
>
> We appreciate your effort and time spent reviewing our submission. We address the weaknesses and questions in the following.
>
> > “Overall, the proposed method consists of several simple heuristics whose justification as a general methodology is not fully clear, and for me, it is somewhat difficult to see significant technical contributions in the paper.”
>
> In HPO research, it is common and unavoidable to use heuristics like expert intuition and domain knowledge. This is rooted in HPO's goal of automating the hyperparameter tuning process traditionally performed by human experts. Due to the NP-hard nature of HPO,  some examples of applying “simple yet effective heuristcs”  in HPO include Hyperband (JMLR’18), BOHB (ICML’18), ASHA (MLSys’20), DyHPO (NeurIPS’22), Hyper-Tune (VLDB’22), PASHA (ICLR’23), DPL (NeurIPS’23). They were all published in top venues in this area,  recognizing the substantial contributions of the heuristics.
>
> Similarly, FastBO also exploits heuristics. The heuristics are carefully designed and are non-trivial, resulting in strong improvement and making substantial technical contributions (the most important contribution is to adaptively identify appropriate fidelities for each configuration in the multi-fidelity setting).
>
>
> > “The model of the learning curve is too simple. I do not agree with that these three functions have enough flexibility to handle a variety of shapes of learning curves. ….”
>
> The functions we chose have shown good fitting and predicting performance in existing empirical learning curve literature. General learning curve shapes can be captured, which is adequate for identifying the efficient and saturation points — The latter is our main focus, while the former serves only as a supporting tool. Our experiments in Section 5.3 demonstrate the effectiveness of our method built on the learning curve modeling.
>
> Given the high efficiency goal of HPO, we simplify the parametric learning curve model to strike a balance between capturing general learning curve shapes and prioritizing computational efficiency. Opting for complex parametric models introduces a computational burden during parameter estimation. The computational burden translates into an increase in the time required for each configuration, and has an adverse effect on the fundamental objective of designing efficient HPO algorithms.
>
> We acknowledge that learning curves can have diverse shapes, sometimes non-convex shapes. However, to the best of our knowledge, existing HPO methods that consider learning curves all exclude the consideration for the poorly-behaved learning curves, particularly those with non-convex shapes. Furthermore, in the SHA-based multi-fidelity methods, there is also an implicit assumption of well-behaved learning curves for hyperparameter configurations. The exclusion of poorly-behaved learning curves is a common practice, mainly due to their complexity and limited impact on the overall optimization process. Having said this, it is straight forward to update the curve fitting model to also include non-convex shapes (but it is believed to be unnecessary in the ML community).
>
> Thanks for your valuable feedback. We have added a separate section in Appendix for an in-depth discussion on the choice of parametric learning curve models.

---

> > ### Author Response · Authors · 2023-11-28
> > **Response (part 2) posted by the chairs on behalf of the authors (no need to respond)**
> >
> > > “Most of well-known BO acquisition functions such as EI, UCB, knowledge gradient, predictive entropy search, and max-value entropy search. A lot of multi-fidelity counter-part has been proposed. The authors ignored most of them.”
> >
> > The focus of our paper is not to introduce new acquisition functions or optimize existing ones. In fact, our method can be applied to any acquisition function. Therefore, we did not include too many acquisition functions in our related work. Considering your comment, we have updated the paper to include knowledge gradient and max-value entropy search in the related work.
> >
> > We would greatly appreciate it if you could specify which particular multi-fidelity methods that you would like us to include. Your feedback is valuable and certainly contributes to improving the paper. We have already covered and discussed relevant multi-fidelity HPO methods in the introduction and related work sections. Additionally, in the experiments, we compared our method with 8 multi-fidelity counterparts in our experiments.
> >
> >
> > > “Appropriate hyper-parameter values for alpha, delta_1, and delta_2 should depend on a given dataset. How can the current setting be justified?”
> >
> > The values for $\alpha$, $delta_1$ and $\delta_2$ are in fact related to the datasets. $\alpha$ is used in warm-up to detect consecutive performance deterioration. Importantly, $\alpha$ is a ratio rather than an absolute value. It represents a proportional performance drop, which reflects our consideration for dataset-specific differences. Regarding $\delta_1$ and $\delta_2$, we already adopted different values for different datasets. We standardized the performance metrics to a scale from 0 to 1 before setting them. Then the values (0.001 and 0.0005) serve as common thresholds for detecting minor performance variations.
> >
> > To avoid introducing unnecessary complexity by tuning a 'hyperparameter' within a 'hyperparameter' optimization algorithm, we intentionally kept a fixed default setting for the hyperparameters on all the datasets. Fine-tuning them is also a possibility and, if explored, may lead to further optimization on performance.

---

### Author Response · Authors · 2023-11-28
**Global response posted by the chairs on behalf of the authors (no need to respond)**

Many thanks to all the reviewers for the thorough review and valuable suggestions. For the whole week before we received the desk rejection, we had been diligently working on addressing all the raised issues and revising the paper incorporating the suggestions. We had prepared a detailed response, expanded the Appendix by 3 pages, and conducted new experiments, which, regrettably, delayed our response.

Through the communication with the PCs, we are deeply grateful [to them for their] willingness to forward our response to the reviewers despite his busy schedule. We will provide a response to each reviewer, although we won't be attaching the revised paper at this time. Thanks to all the reviewers again for their understanding.